# Clinical Implications of Glyoxalase1 Gene Polymorphism and Elevated Levels of the Reactive Metabolite Methylglyoxal in the Susceptibility of Type 2 Diabetes Mellitus in the Patients from Asir and Tabuk Regions of Saudi Arabia

**DOI:** 10.3390/jpm12040639

**Published:** 2022-04-15

**Authors:** Muhanad Alhujaily, Mohammad Muzaffar Mir, Rashid Mir, Mushabab Ayed Abdullah Alghamdi, Javed Iqbal Wani, Zia ul Sabah, Imadeldin Elfaki, Tarig Mohammad Saad Alnour, Mohammed Jeelani, Mosleh Mohammad Abomughaid, Samir Abdulkarim Alharbi

**Affiliations:** 1Department of Clinical Laboratory Sciences, College of Applied Medical Sciences, University of Bisha, Bisha 61922, Saudi Arabia; malhujaily@ub.edu.sa (M.A.); moslehali@ub.edu.sa (M.M.A.); 2Department of Basic Medical Sciences, College of Medicine, University of Bisha, Bisha 61922, Saudi Arabia; mjeelani@ub.edu.sa; 3Prince Fahd Bin Sultan Research Chair, Department of MLT, Faculty of Applied Medical Sciences, University of Tabuk, Tabuk 71491, Saudi Arabia; rashid@ut.edu.sa (R.M.); telnour@ut.edu.sa (T.M.S.A.); 4Department of Internal Medicine, College of Medicine, University of Bisha, Bisha 61922, Saudi Arabia; mualghamdi@ub.edu.sa; 5Department of Internal Medicine, College of Medicine, King Khalid University, Abha 61421, Saudi Arabia; drjiwani1959@gmail.com (J.I.W.); drziaulsabah@gmail.com (Z.u.S.); 6Department of Biochemistry, Faculty of Science, University of Tabuk, Tabuk 71491, Saudi Arabia; ielfaki@ut.edu.sa; 7Department of Medical Laboratory Sciences, College of Applied Medical Sciences, Shaqra University, Shaqra 11961, Saudi Arabia; saalharbi@su.edu.sa

**Keywords:** T2DM, methylglyoxal, GLO1-glyoxalase-1, single nucleotide polymorphism, rs2736654 C>A, rs1130534 T>A, type 2 diabetes in Saudi Arabia

## Abstract

Diabetes mellitus constitutes a big challenge to the global health care system due to its socioeconomic impacts and very serious complications. The incidence and the prevalence rate are increased in the Gulf region including the KSA. Type 2 diabetes mellitus (T2DM) is caused by diverse risk factors including obesity, unhealthy dietary habits, physical inactivity, smoking and genetic factors. The molecular genetic studies have helped in the detection of many single nucleotide polymorphisms (SNP) with different diseases including cancers, cardiovascular diseases and T2DM. The glyoxalase 1 (GLO1) is a detoxifying enzyme and catalyzes the elimination of the cytotoxic product methylglyoxal (MG) by converting it to D-lactate, which is not toxic to tissues. MG accumulation is associated with the pathogenesis of different diseases including T2DM. In this study, we have investigated the association of the glyoxalase 1 SNPs (rs2736654) rs4746 C>A and rs1130534 T>A with T2DM using the amplification refractory mutation system PCR. We also measured the concentration of MG by ELISA in T2DM patients and matched heathy controls. Results show that the CA genotype of the GLO rs4647 A>C was associated with T2DM with OR = 2.57, *p*-value 0.0008 and the C allele was also associated with increased risk to T2DM with OR = 2.24, *p*-value = 0.0001. It was also observed that AT genotype of the rs1130534 was associated with decreased susceptibility to T2DM with OR = 0.3, *p*-value = 0.02. The A allele of rs1130534 was also associated with reduced risk to T2DM with PR = 0.27 = 0.006. In addition, our ELISA results demonstrate significantly increased MG concentrations in serum of the T2DM patients. We conclude that the GLO1 SNP may be associated with decreased enzyme activity and a resultant susceptibility to T2DM. Further well-designed studies in different and large patient populations are recommended to verify these findings.

## 1. Introduction

A ubiquitously expressed and highly conserved glyoxalase enzyme network plays a very crucial biological role by detoxifying methylglyoxal (MG) and other endogenous toxic metabolites to harmless substances such as D-lactate [1,2,3]. Also known as the glyoxalase enzyme system, it consists of two enzymes, glyoxalase 1 (EC4.4.1.5, GLO1) and glyoxalase-2 (EC 3.1.2.6, GLO2) and uses reduced glutathione (GSH) [4,5,6]. A highly reactive dicarbonyl compound, MG is formed from carbohydrates, proteins and lipids and mainly from the non-enzymatic reactions with glyceraldehyde-3-phosphate and dihydroxy acetone phosphate in the glycolytic pathway [4,7]. The main targets of MG are proteins and nucleic acids resulting in a vast array of advanced glycation end products (AGEs), DNA adducts, DNA strand breaks, mutagenicity and cytotoxicity [8,9,10]. The increased levels of MG and AGEs have been implicated in many disease processes including diabetes, obesity [11], cardiovascular diseases [8,12], hypertension [8,13], cerebrovascular diseases [14,15], Parkinson’s disease [8,16] aging [8,17] and cancer [18]. A graphic representation of the Glyoxalase/methylglyoxal system is shown in Figure 1.

Diabetes mellitus (DM) is one of the major global problems and the kingdom of Saudi Arabia (KSA) too has a high prevalence [19]. In general, there are two types of DM. Type 1 DM is caused by the destruction of the pancreatic beta cells that secrete the insulin [20], and Type 2 DM (T2DM) which develops by tissue resistance to insulin action and pancreatic beta cells dysfunction [21]. DM is associated with acute consequences like diabetic ketoacidosis, hyperosmolar hyperglycemic syndrome and chronic complications such as renal failure, blindness, cardiovascular disease and diabetic neuropathy [22]. These complications, unfortunately but invariably, result in high rates of morbidity and mortality. Both T1DM and T2DM are heterogeneous and polygenic in nature with distinct characteristics [19,23].

The glyoxalase system is an important and integral part of the metabolic network in humans and plays a crucial role in neutralizing the deleterious effects of MG. Although GLO1 and GLO2 function as a system, the role of GLO1 in health and disease processes has been more extensively studied. GLO1 is a metal-dependent dimeric enzyme and each of its subunits contains one Zn^2+^ [4]. The decreased activity/expression of GLO1 has been reported in diabetes [8,24] neurodegenerative disorders [16,25], cardiovascular diseases [12,24] and cerebrovascular diseases [15,26]. In addition, the reduced activity and expression of GLO1 have also been observed in aging [17,27], epigenetic changes [8], male infertility [28], female infertility [29], multidrug resistance and malignancies [30]. Many researchers have reported the competing roles of MG, AGEs and GLO1 in the pathogenesis of diabetes and diabetic complications such as retinopathy, nephropathy and neuropathy [31].

The human GLO1 gene is located on chromosome 6 at a diallelic locus 6p21.2 between HLA and centromere [6,32]. Many studies have reported the effects of GLO1 gene expression and posttranslational modifications of GLO1 on its activity [8,33,34]. The single nucleotide polymorphism (SNP) in GLO1 gene has been studied by different groups with varying results [35,36]. Peculis et al. [37] reported an association of T allele of rs10449346 and T allele of rs1130534 with decreased GLO activity, being more manifest in subjects with haplotypes of these two SNPs. Wu et al. reported a positive association of rs1049346 with diabetic complications including retinopathy and neuropathy but no association was established between rs2736654 and diabetic complications [38]. High fasting levels of MG have been associated with SNP rs13199033 located in the 3′ UTR (untranslated region) of GLO1 [39]. The reports on the association of SNPs in the GLO1 gene with the markers of the glyoxalase pathway are inconsistent and need further investigations. To the best of our knowledge, we have not come across any such report in the Kingdom of Saudi Arabia (KSA). In the current study, we have investigated the association of glyoxalase 1 SNPs rs2736654 C>A and rs1130534 T>A with T2DM. We also measured the concentration of the substrate of MG, the glyoxalase 1, in T2DM patients in the Asir and Tabuk regions of Saudi Arabia. The location of Asir and Tabuk regions in Saudi Arabia is highlighted in Figure 2.

## 2. Methodology

### 2.1. Study Population

This population-based case control and a collaborative study was conducted on 101 T2DM patients and matched healthy controls (110–144). Specimens were collected from the Asir and Tabuk regions of Saudi Arabia. Informed consent was obtained before collecting samples from all patients and control subjects.

### 2.2. Ethical Approval

The ethical approval was obtained from the local RELOC committee in the College of Medicine, University of Bisha (Ref. No. UBCOM/H-06-BH-087(08/32), in accordance with the principles of the Helsinki Declaration. Informed consent was obtained before collecting samples from all patients and control subjects.

### 2.3. Selection of Study Population

This case-control study enrolled 101 subjects with type 2 diabetes mellitus (T2DM) and about 150 normal control subjects. T2DM diagnosis was based on the WHO criteria.

#### 2.3.1. Inclusion Criteria for Patients and Controls

The patient group included confirmed patients of T2DM, visiting the hospital in the Asir and Tabuk regions of Saudi Arabia for routine checkups and follow-ups.

The control subjects were matched healthy volunteers with no history of diabetes or any major clinical disorders and had normal fasting plasma glucose levels.

#### 2.3.2. Exclusion Criteria

The T1DM patients and T2DM patients with other significant chronic diseases or malignancies were excluded from the study.

#### 2.3.3. Data Collection

The variables that were analyzed from the study subjects included the case history, age, gender, BMI, duration of T2DM, glycated hemoglobin (HbA1c), fasting glucose, random blood glucose, total cholesterol, triacylglycerol, high-density lipoprotein-cholesterol (HDL-C), and low-density lipoprotein cholesterol (LDL-C) concentrations and total cholesterol/HDL-C ratios. The biochemical parameters were assayed using the standard protocols on a fully automatic analyzer (Cobas Integra 800, Roche, Basel, Switzerland).

### 2.4. Sample Collection from T2DM Patients

A total of 3 mL of peripheral blood sample was collected in an EDTA or Lavender top tube from all T2DM patients. The blood specimens were immediately stored at −20 °C to −30 °C until further molecular studies. Another aliquot of blood (about 2 mL) was collected in a red top tube and immediately sent for biochemical analyses.

### 2.5. Sample Collection from Healthy Controls

All healthy age matched controls specimens were timed around blood draws that were part of routine investigations and did not require additional phlebotomy. Written and informed consent was obtained from all the participating subjects before the sample collection. A 3 mL peripheral blood sample was collected in an EDTA or lavender top tube from all controls. The blood specimens were immediately stored at −20 °C to −30 °C until further molecular studies. Another aliquot of blood (about 2 mL) was collected in a red top tube and immediately sent for biochemical analyses.

### 2.6. DNA Extraction from T2DM Patients

Genomic DNA was isolated using the ThermoScientific Genomic DNA purification kit (Thermo Scientific, Waltham, MA, USA) from the whole blood according to the manufacturer’s instructions. The DNA integrity was checked with agarose gel electrophoresis and NanoDrop (Thermo Scientific, Waltham, MA, USA).

### 2.7. Genotyping of GLO-I rs2736654 C>A, rs1130534 T>A

The genotyping of *GLO-I rs2736654* C>A was done by Allele Specific PCR and *GLO-I* rs1130534 T>A genotyping was established by Amplification-Refractory Mutation System-PCR (ARMS-PCR) and primers were designed by using primer3 software as depicted in Table 1.

### 2.8. Preparation of PCR Cocktail

The ARMS-PCR was done in a reaction volume of 25 µL containing template DNA (50 ng), F1—0.12 µL, R1—0.12 µL, F2—0.12 µL, R2—0.12 µL of 25 pmol of each primer and 5 µL from GoTaq^®^ Green Master Mix (Cat M7122) (Promega, Woods Hollow Road·Madison, WI, USA). The final volume of 25 µL was adjusted by adding nuclease free double distilled water. Finally, 2 µL of DNA was added from each patient. The thermocycling conditions used were at 95 °C for 10 min followed by 40 cycles of 95 °C for 35 s, annealing temperature GLO1-rs2736654 C>A for 60 °C and rs1130534 T>A for 62 °C, extension at 72 °C for 35 s and final extension at 72 °C for 10 min. PCR products were separated on 2% agarose gel stained with 2 µL of sybre safe stain and visualized on a UV transilluminator from Bio-Rad (Hercules, CA, USA).

For GLOI- rs2736654 SNP, primers F1 and R2 amplified a wild-type allele (C allele), generating a band of 403 bp and primers F2 and R1 generate a band of 178 bp from the mutant allele (A allele) as shown in Figure 3.

In the case of GLOI -rs1130534 T>A SNP, Primers F1 and R1 flank the exon of the *GLOI* -rs1130534 T>A gene, resulting in a band of 456 bp to act as a control for DNA quality and quantity. Primers F1 and R2 amplified a wild-type allele (A allele), generating a band of 336 bp and primers F2 and R1 generated a band of 213 bp from the mutant allele (A allele) as depicted in Figure 4.

### 2.9. Estimation of Methylglyoxal

The methylglyoxal concentrations were determined by ELSA kits supplied by Abbkine, Inc., (Wuhan, China) by following the manufacturer’s instructions. In brief, stock standard (960 pg/mL) was 2-fold diluted to prepare the standard curve. A total of 40 µL of sample diluent was pipetted into all wells except the wells of blank and STDs; 50 µL of duplicate STDs and 10 µL of serum samples were pipetted into their corresponding ELISA wells. The plate was mixed and incubated at 37 °C for 45 min. The plate was washed 5 times with washing buffer and 50 µL conjugate was added to all wells except blanks and incubated at 37 °C for 30 min. The plate was washed 5 times, and 50 µL of chromogenic solution A and 50 µL of chromogenic solution B were added into all wells. The plate was again incubated at 37 °C in the dark for 15 min. a total of 50 µL of the Stop solution was added to all wells and the plate O.D was measured at 450 nm using ELISA reader (Human).

### 2.10. Statistical Analysis

Chi-square test was used to evaluate the differences in the *GLOI-rs2736654* and *rs2736654* allele and genotype frequencies between study groups. The associations between *GLOI rs2736654* and *rs2736654* genotypes and risk ratio were estimated by computing the odds ratios (ORs), risk ratios (RRs) and risk differences (RDs) with 95% confidence intervals (CIs). Allele frequencies among patients as well as controls were evaluated by using the Chi–square Hardy-Weinberg equilibrium test. A *p*-value < 0.05 was considered significant. All statistical analyses were performed using Graph Pad Prism 6.0 or SPSS 16.0. Expression deregulation of serum methylglyoxal levels were correlated with the T2DM patients using Graph Pad Prism 6.0 or SPSS 16.0.

## 3. Results

### 3.1. Demographic Features and Baseline

The demographic features and baseline characteristics of T2DM patients are summarized in Table 2. Only details of 101 patients for whom we obtained discrete results on electrophoresis are included. Of 101 consecutive patients, 52 were males, 49 were females, 41 patients were below 40 years and 60 were equal to or above 40 years of age. In the T2DM patients, 14 had fasting glucose ≤110 mg/dL and 87 were having fasting glucose >110 mg/dL. In total, 60 T2DM patients had total cholesterol ≤200 and 41 had total cholesterol >200 mg/dL. Among 101 T2DM patients, 64 had triglycerides greater than <150 mg/dL and 37 had triglycerides equal to or lesser than 150 mg/dL. HbA1c was >6% in 70 T2DM patients and ≤6% in 31 patients. A comparison of the clinical and biochemical features of T2DM patients and controls is summarized in Table 3. Differences in the mean of the blood glucose fasting/random, HDL-C, LDL-C and total cholesterol were significant between the patient and controls as is evident in Table 3.

### 3.2. Statistical Comparisons between T2DM Patients and Controls for *GLO-I rs2736654* C>A Genotypes

At the time of analysis, 100 T2DM patients displayed discrete results in the gel electrophoresis whereas only 144 healthy controls displayed sharp bands in the gel electrophoresis, so we present the results for 100 patients and 144 controls. Our results indicate that there were significant differences in genotype distribution of the GLO rs4647 A>C or *GLO-I rs2736654* A>C genotypes between T2DM patients and controls (*p* < 0.0001) (Table 4). The frequencies of AA, AC and CC genotypes in patients are 30%, 58%, and 12%, respectively, and in controls are 55.6%, 41.7%, 2.7%, respectively. In the allelic comparison, a significantly higher frequency of C allele (0.41) was reported in T2DM patients than the healthy controls (0.24) *p* = 0.0001.

### 3.3. Estimation of Association between GLO-I rs2736654 C>A Gene Variation with T2DM by Multivariate Analysis

An unconditional logistic regression was used to estimate associations between the genotypes and the risk of T2DM patients (Table 5). Logistic regression analysis was done to determine the adjusted odds ratios (OR) and 95% confidence intervals (95% CI) associated with the risk of T2DM, after controlling for several covariates, taking control women as the reference group. It was found that an increased risk of T2DM was associated with the Glyoxalase I gene polymorphism genotype in an allele dosage-dependent manner. Our results indicate that in the codominant model, the CA genotype of the glyoxalase I gene is associated with increased susceptibility to T2DM with OR 2.57 (95%) CI = (1.4821 to 4.4835), RR = 1.43 (1.15–1.76), *p* < 0.0008 (Table 5). Also CC genotype of the glyoxalase I gene is associated with increased susceptibility to T2DM with OR 8.57 (95%) CI (2.3930 to 26.744), RR = 2.90 (1.23–2.84), *p* < 0.0007. A significant association is observed in the case of dominant as well as in recessive inheritance models. While determining the allelic comparisons, the C allele of glyoxalase I gene polymorphism is strongly associated with increased susceptibility to T2DM disease with OR = 2.24(95%) CI (1.519–3.326), RR = 1.43(1.184–1.740), *p* < 0.0001).

### 3.4. Statistical Comparisons of GLO-I rs2736654 C>A (rs4647 A>C) Genotypes with Demographic Features and Biochemical Characteristics T2DM Patients

The statistical comparisons (*p*-values) of GLOI-rs4647 A>C genotypes with demographic features and biochemical characteristics in T2DM patients was done by using a multivariate analysis based on logistic regression like odds ratio (OD) and risk ratio (RR) with 95% confidence intervals (CI) (Table 6). Results show that there was a strong correlation between the GLOI-rs4647 A>C genotypes with respect to the RBS, Total cholesterol, LDL-C, TG and HbA1c.

### 3.5. Statistical Comparisons of GLO-rs1130534 T>A between T2DM Patients and Controls

At the time of analysis, all the 101 T2DM patients display results in the gel electrophoresis for GLO-rs1130534 T>A, whereas only 100 healthy controls display sharp bands in the gel, so we present here results for only 100 controls. Our results indicate that there was a significant difference in the distribution of the GLO-rs1130534 T>A genotypes between T2DM patients and controls (*p* < 0.048) as can be seen in Table 7. The frequency of genotypes TT/AT/AA in T2DM patients is 89.10%, 7.92% and 2.97%, respectively, and in controls is 80%, 19%, 1%), respectively. A higher frequency of T allele (0.97) is observed in T2DM patients than in the healthy controls (0.90).

### 3.6. Multivariate Analysis to Estimate the Association between GLO-rs1130534 T>A Gene Variation in T2DM Patients

Our results indicate that the AT genotype of the rs1130534 was associated with decreased susceptibility to T2DM with OR = 0.3(0.1554 to 0.901), RR = 0.66(0.4993 to 0.89), *p*-value = 0.02 (Table 8). Similarly, our results indicate that the A allele of the GLO-rs1130534 was associated with reduced risk or decreased susceptibility to T2DM with OR = 0.27(0.1073 to 0.689), RR = 0.62(0.4996 to 0.78), *p*-value = 0.006 (Table 8), therefore the A allele was protective in T2DM patients (Table 8).

### 3.7. Statistical Comparisons between GLO-rs1130534 T>A Genotypes and T2DM Patient Characteristics

The statistical comparisons (*p*-values) of GLO-1 rs1130534 T>A genotypes with demographic and biochemical features of T2DM were done by using a multivariate analysis model based on logistic regression such as odds ratio (OD) and risk ratio (RR) with 95% confidence intervals (CI) (Table 9). Results show that there was a strong correlation between the GLO-rs1130534 T>A genotypes with respect to the fasting blood glucose, total cholesterol, HDL-C, LDL-C and HbA1c.

### 3.8. Correlation between Serum Methylglyoxal Levels and T2DM

The methylglyoxal concentrations were estimated in 80 T2DM patients and 80 healthy controls and the results are summarized in Table 10A. Overall, the mean serum methylglyoxal concentration in T2DM patients was 258.87 ± 77.10 pg/mL, which is significantly higher as compared to controls (Table 10A). There is no significant difference in the serum methylglyoxal levels in male and female patients (Table 10B). The serum methylglyoxal is significantly high in the age group of > 40 years compared to the age group of ≤40 years in patients. (Table 10C).

## 4. Discussion

T2DM represents a huge health challenge all over the world including KSA due to its socioeconomic effects, and very serious complications [19,40]. T2DM is caused by the impaired peripheral insulin action in the liver, muscles and adipose tissues as well as by pancreatic beta cell demise [20,41]. The risk factors for T2DM are multiple and include obesity, unhealthy diets, physical inactivity, genetic and others [19,20,21,22,23]. The human glyoxalase system present in cytosol or nucleus is responsible for the detoxification of a highly reactive dicarbonyl compound, MG, that is produced from the metabolism of carbohydrates, lipids and proteins, but mainly from the carbohydrate metabolism [1,2,3]. The accumulated MG and AGEs is involved in the development of T2DM, CVDs and other diseases [11,12,13,14,15,16,17,18]. The complex and intricate roles of MG, AGEs and GLO1 in the pathogenesis of diabetes and diabetic complications such as retinopathy, nephropathy and neuropathy has also been reported [31]. A positive association of rs1049346 in GLO1 with diabetic complications including retinopathy and neuropathy has also been reported but no association was established between rs2736654 and diabetic complications [38]. High fasting levels of MG have been associated with SNP rs13199033 located in the 3′ UTR (untranslated region) of the GLO1 [39]. The reports on the association of SNPs in GLO1 gene with the markers of the glyoxalase pathway are inconsistent and mostly inconclusive. In animal models, it has been shown that MG induces glycation, which impairs the angiogenesis in fat tissues and worsens the insulin sensitivity that may lead to hypoxia and the development of T2DM [42].

Our results show that there was a significant difference in rs2736654 (rs4647) A>C SNP genotype distribution between patients and controls (*p*-value > 0.05, Table 4), and that the CA genotype and the C allele were associated with increased susceptibility to T2DM (Table 5). Furthermore, the results show that the rs2736654 (rs4647) A>C genotype distribution is significantly different between patients with normal and patients with increased fasting blood glucose and HbA1c (Table 6). The rs2736654 (rs4647) A>C changes Glutamic acid to Alanine [43]. At physiologic pH, Glutamic acid is negatively charged and alanine is uncharged. Therefore, the rs4647 A>C may have a profound impact on GLO structure and function. Our results show that CA genotype and the C allele were associated with increased susceptibility to T2DM (Table 5) which is probably due to reduced GLO1 enzyme activity (as a result of structural perturbation) that leads to the accumulation of its cytotoxic substrate MG, hereby causing insulin resistance and T2DM [44,45]. This postulation is substantiated by our ELISA data that show significantly increased concentrations of the MG in patients as compared to the control group (Table 10A). However, more molecular genetic studies on larger sample sizes in different populations are needed to explain the exact phenomena in detail. Results also show that there was no significant difference in the MG concentration between male and female patients (Table 10B). Our ELISA results also show that there was a significantly elevated MG concentration in elder patients ≥40 years old) than in younger ones (<40 years old) (Table 10C).

Our result is also in agreement with a study that showed the implication of the GLO1 and its substrate the MG in diabetes [24], and also another study that reported that single nucleotide variations in the GLO may induce diabetes and diabetic complications [37]. It was also observed that the rs4647 A>C genotype distribution is significantly different in patients with elevated lipid profiles and patients with normal lipid profiles (Table 6). This result is in agreement with previous reports that suggested that the accumulation of MG (the substrate of GLO1) leads to the development of diabetic cardiovascular complications [12,46].

Our results also show that the AT genotype of GLO-1 rs1130534 T>A and A allele were associated with decreased susceptibility to T2DM (Table 8). There were significant differences rs1130534 T>A in patients with normal blood sugar and patients with increased blood sugars, and patients with normal and patients with abnormal lipid profiles (Table 9). It has been reported that rs1130534 SNP causes a synonymous replacement at codon 124 (GGA to GGT) and as a result, amino acid Glycine is not changed [47]. However, our result is inconsistent with a study that reported that A allele is associated with reduced GLO1 enzyme activity [43]; this inconsistency is probably because of different patient ethnicities, sample sizes or other factors. We do not have an explanation of why the rs1130534 SNP was associated with T2DM although there is no change in amino acid (Glycine-124-Glycine); however, this result may be partially consistent with a study that demonstrated that rs1130534 SNP affects GLO1 enzyme activity [37]. Our observations need to be further validated in future studies.

## 5. Conclusions

In summary, we have investigated the association of glyoxalase 1 SNPs rs2736654 C>A and rs1130534 T>A with T2DM in patients from Asir and Tabuk regions of Saudi Arabia. It is concluded that the GLO1 C332T (Ala111Glu) rs4746 was strongly associated with increased susceptibility to T2DM with OR 2.57 (95%) CI (1.4821 to 4.4835), RR = 1.43 (1.15–1.76), *p* < 0.0008. A significant association is observed in dominant as well as in recessive inheritance models. Similarly, GLO1 C332 (Ala111) allele is strongly associated with increased susceptibility to T2DM disease with OR = 2.24 (95%) CI (1.519–3.326), RR = 1.43(1.184–1.740), *p* < 0.0001). In addition, our ELISA result demonstrated significantly increased methylglyoxal concentrations in serum of the T2DM patients as compared to the control group. The limitations of our study include a smaller study population with restricted study criteria. Future molecular studies on these aspects on larger patient populations and in different ethnic groups are recommended.

## Figures and Tables

**Figure 1 jpm-12-00639-f001:**
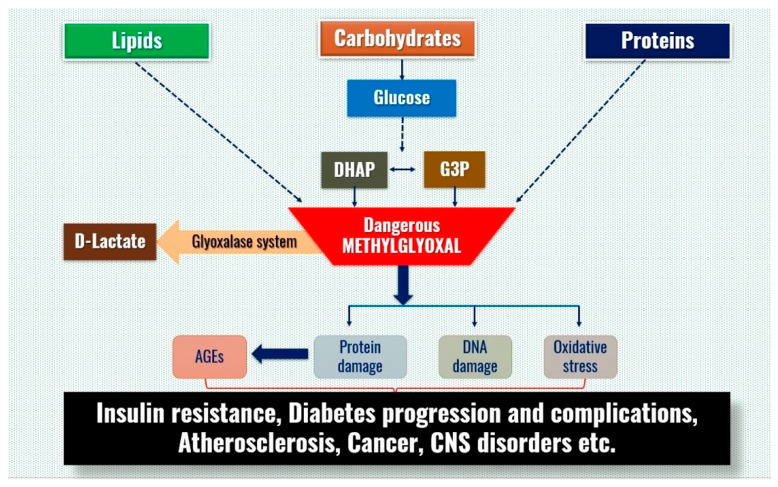
Schematic representation of Glyoxalase/methylglyoxal system. The cytotoxic methylglyoxal is rendered harmless after conversion to D-lactate by Glo-1. Any imbalance in this system leads to intracellular and extracellular damage.

**Figure 2 jpm-12-00639-f002:**
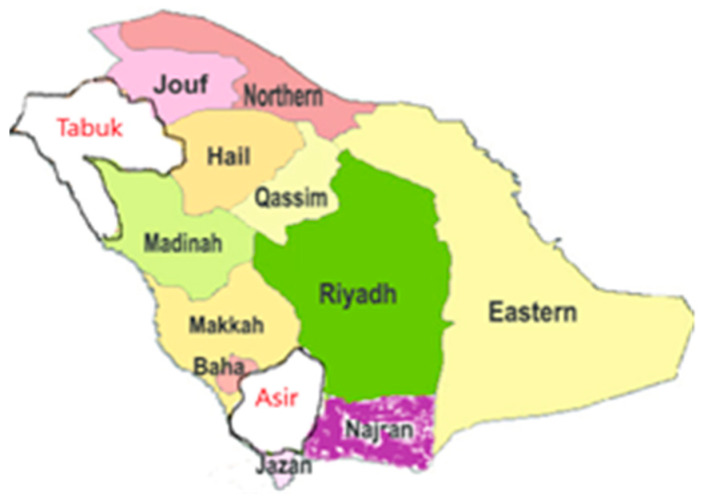
Region wise map of Saudi Arabia. The two study regions of Tabuk and Asir are labelled with red color.

**Figure 3 jpm-12-00639-f003:**
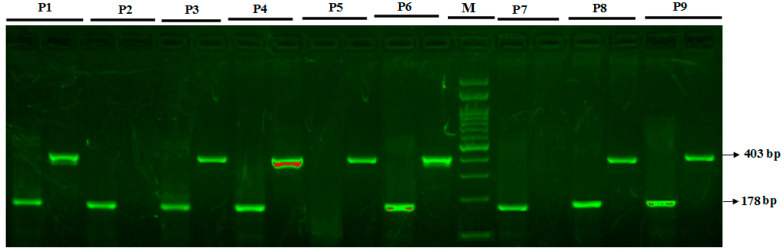
Detection of glyoxalase 1 genotyping, GLO-I rs2736654 C>A primers utilizing Allele Specific PCR) in T2DM patients. M-100 bp DNA ladder; Heterozygous CA—in patients P1, P3, P4, P6, P8, P9; Homozygous CC—P5; Homozygous A/A—P2, P7.

**Figure 4 jpm-12-00639-f004:**
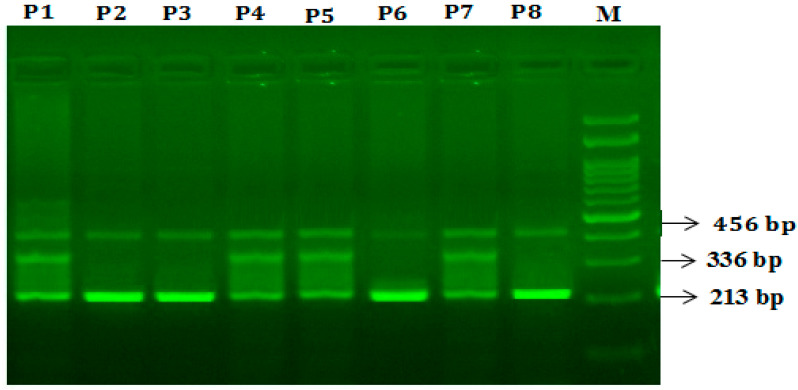
Detection of glyoxalase 1 genotyping *GLOI* rs1130534 T>A utilizing Refractory Mutation System PCR (ARMS-PCR) in T2DM patients. M-100 bp DNA ladder; Heterozygous A/T—in patients P1, P4, P5, P7; Homozygous T/T—in patients P2, P3, P6, P8.

**Table 1 jpm-12-00639-t001:** Allele-Specific PCR (AS-PCR) primers for GLO1 SNPs.

GLO-I rs2736654 C/A (E>A) Primers			
Gene	Sequence	Annealing Tempt	PCR Product
GLO-I F1	5′-GAGAGACTGGAGATCAAGGCAG-3′	60 °C	403 bp
GLO-I R2-C	5′-CAATTGGGGCACTGAAGATGATGC-3′		
GLO-I F2-A	5′-CCATTGTGGTAACTCTGGGTCT-3′		
GLO-I R1	5′-TTTTTGTAGCAGGGGTTAGGCCA-3′		178 bp
GLO-I rs1130534 T>A primers			
GLO-I F1	5′-GTGTATGTGCTAGCAGAAACTGG-3′	62 °C	456 bp
GLO-I R1	5′-ATGCAGGGGTTAGGCCAATTATG-3′		
GLO-I F2 T	5′-CATAAAAACAGGCAAACTTACCGAAT-3′		213 bp
GLO-IR2 A	5′-GGCAATTCAGACCCTCGAGGT-3′		336 bp

**Table 2 jpm-12-00639-t002:** Demographic features and baseline characteristics of T2DM patients.

Subjects	*n* = 101	Percentage
Gender distribution		
Males	52	51.48%
Females	49	48.51%
Age distribution		
Age < 40	41	40.59%
Age ≥ 40	60	59.40%
Fasting blood glucose		
Glucose ≤ 110 mg/dL	14	13.86%
Glucose > 110 mg/dL	87	86.13%
Random blood glucose		
RBS ≤ 200 mg/dL	56	55.44%
RBS > 200 mg/dL	45	44.55%
Total cholesterol		
Cholesterol ≤ 200 mg/dL	60	59.40%
Cholesterol > 200 mg/dL	41	40.59%
HDL-C		
HDL-C ≤ 55 mg/dL	73	72.25
HDL-C > 55 mg/dL	28	25.74
LDL-C		
LDL ≤ 100 mg/dL	75	74.25%
LDL > 100 mg/dL	26	25.74%
TG		
TG ≤ 150 mg/dL	64	63.36%
TG > 150 mg/dL	37	36.63
HbA1c		
HbA1c ≤ 6%	70	69.30%
HbA1c > 6%	31	30.69%
Creatinine		
Creatinine ≤ 1.35 mg/dL	83	82.17%
Creatinine > 1.35 mg/dL	18	17.82%

RBS—Random blood sugar, HbA1c—glycated hemoglobin, TG—Triglycerides, LDL-C—low-density lipoprotein Cholesterol, HDL-C—High-density lipoprotein cholesterol.

**Table 3 jpm-12-00639-t003:** Comparison of clinical and biochemical characteristics of T2DM patients and controls.

Characteristic	Controls ^a^	Cases ^a^	*p* ^b^
Age	30.32 ± 5.6.8	29.30 ± 4.88	0.357
HbA1c	5.5 ± 0.17	5.90 ± 0.43	0.122
Fasting blood glucose	95.4 ± 1.5	120.56 ± 2.33	<0.001
BMI (kg/m^2^) ^c^	24.83 ± 2.45	28.5 ± 2.64	<0.001
Cholesterol (mg/dL) ^c^	100.5 ± 1.17	230.80 ± 5.58	<0.001
TG (mg/dL) ^c^	70.85 ± 1.8	105.81 ± 2.39	0.066
HDL (mg/dL) ^c^	60.87 ± 3.9	30.92 ± 3.9	<0.001
LDL (mg/dL) ^c^	100.07 ± 2.67	150.63 ± 3.98	<0.002

^a^ 101 cases and 144 controls. ^b^ Student’s t-test for continuous variables (variables with normal distribution), ^c^ Values as mean ± standard deviation, Body mass Index (BMI), Triglycerides (TG), High density lipoprotein cholesterol (HDL-C), Low density lipoprotein cholesterol (HDL-C).

**Table 4 jpm-12-00639-t004:** GLO-I rs2736654 C>A (rs4647 A>C) genotype distribution in patients and in controls.

	*n*	AA	AC	CC	Df	χ^2^	A	C	*p* Value
T2DM patients	100	30(30%)	58(58%)	12(12%)	02	19.46	0.59	0.41	0.0001
Controls	144	80(55.6%)	60(41.7%)	04(2.7%)			0.76	0.24	

**Table 5 jpm-12-00639-t005:** Association of *GLO-I rs2736654* C>A (GLO rs4647 A>C) gene variation with T2DM.

Mode of Inheritance	Controls (*n* = 144)	T2DM Patients (*n* = 100)	OR (95% CI)	RR (95% CI)	*p*-Value
Co-dominant model					
GLO-AA	80	30	1 (ref.)	1 (ref.)	
GLO-CA	60	58	2.57(1.4821 to 4.4835)	1.43(1.158 to 1.76)	0.0008
GLO-CC	04	12	8.0(2.3930 to 26.744)	2.90(1.235 to 6.84)	0.0007
Dominant model					
GLO-AA	80	30	1 (ref.)	1 (ref.)	
GLO (CA + CC)	64	70	2.91(1.7007 to 5.002)	1.52(1.2333 to 1.88)	0.0001
Recessive model					
GLO (AA + CA)	140	88	1 (ref.)	1 (ref.)	
GLO-CC	04	12	4.77(1.492 to 15.264)	2.45(1.04 to 5.77)	0.0084
Allelic comparison					
GLO-A	220	118	1 (ref.)	1 (ref.)	
GLO-C	68	82	2.24(1.519 to 3.326)	1.43(1.184 to 1.740)	0.0001

OR = Odds Ratio, RR = Risk Ratio, CI = Confidence interval.

**Table 6 jpm-12-00639-t006:** Associations of GLO I-rs4647 A>C genotypes with patient characteristics.

Subjects	*n* = 100 *	TT	TA	AA	χ^2^	df	*p* Value
Association with gender							
Males	51	18	30	3	4.23	2	0.12
Females	49	12	28	9			
Association with Age							
Age < 40	40	16	21	3	3.69	2	0.158
Age ≥ 40	60	14	37	9			
Fasting glucose							
Glucose ≤ 110 mg/dL	14	6	6	2	1.61	2	0.44
Glucose > 110 mg/dL	86	24	52	10			
Association with RBS							
RBS ≤ 200 mg/dL	56	10	40	6	10.39	2	0.0058
RBS > 200 mg/dL	44	20	18	6			
Association with Cholesterol							
Cholesterol ≤ 200 mg/dL	60	10	44	6	15.47	2	0.0004
Cholesterol > 200 mg/dL	40	20	14	6			
Association with HDL-C							
HDL-C ≤ 55 mg/dL	75	20	46	9	1.69	2	0.42
HDL-C > 55 mg/dL	25	10	12	3			
Association with LDL-C							
LDL ≤ 100 mg/dL	74	15	50	9	13.48	2	0.0012
LDL > 100 mg/dL	26	15	8	3			
Association with TG							
TG ≤ 150 mg/dL	64	18	43	3	10.72	2	0.004
TG > 150 mg/dL	36	12	15	9			
Association with HBA1c%							
HBA1c ≤ 6%	30	16	8	6	17.03	2	0.0002
HBA1c > 6%	70	14	50	6			
Association with Creatinine							
Creatinine ≤ 1.35 mg/dL	82	23	50	9	4.68	2	0.09
Creatinine > 1.35 mg/dL	18	7	8	3			

* Out of 101 T2DM patient samples, one sample did not give good gel bands for GLO-I rs2736654 and as such we included only 100 samples for analyses. RBS—Random blood sugar, HbA1c—glycated hemoglobin, TG—Triglycerides, LDL-C—low-density lipoprotein Cholesterol, HDL-C—High-density lipoprotein cholesterol.

**Table 7 jpm-12-00639-t007:** GLO-rs1130534 T>A SNP genotypes distribution in patients and controls.

	*n*	TT%	AT%	AA%	Df	χ^2^	T	A	*p* Value
T2DM patients	101 *	90(89.10)	08(7.92)	03(2.97)	2	6.06	0.97	0.7	0.048
Controls	100 *	80(80)	19(19)	01(1)			0.90	0.10	

* At the time of analysis, all the 101 T2DM patients display results in the gel electrophoresis for GLO-rs1130534 T>A, whereas only 100 healthy controls display sharp bands in the gel, so we present here results for only 100 controls.

**Table 8 jpm-12-00639-t008:** Association between GLO-rs1130534 T>A gene variations with T2DM.

Mode of Inheritance	Controls (*n* = 100)	T2DM Patients (*n* = 101)	OR (95% CI)	RR (95% CI)	*p*-Value
Co-dominant model					
GLO-TT	80	90	1 (ref.)	1 (ref.)	
GLO-AT	19	08	0.3(0.1554 to 0.901)	0.66(0.4993 to 0.89)	0.02
GLO-AA	01	03	2.66(0.2719 to 26.153)	1.88(0.342 to 10.35)	0.39
Dominant model					
GLO-TT	80	90	1 (ref.)	1 (ref.)	
GLO (AT-AA)	20	11	0.48(0.2208 to 1.082)	0.72(0.537 to 0.99)	0.07
Recessive model					
GLO (TT-AT)	99	98	1 (ref.)	1 (ref.)	
GLO-AA	01	03	3.03(0.309 to 29.640)	2.01(0.366 to 11.03)	0.34
Allelic Comparison					
GLO-T	79	95	1 (ref.)	1 (ref.)	
GLO-A	21	06	0.27(0.1073 to 0.689)	0.62(0.4996 to 0.78)	0.006

At the time of analysis, all the 101 T2DM patients display results in the gel electrophoresis for GLO-rs1130534 T>A, whereas only 100 healthy controls display sharp bands in the gel, so we present here results for only 100 controls. OR = Odds Ratio, RR = Risk Ratio, CI = Confidence interval.

**Table 9 jpm-12-00639-t009:** Association of GLO-rs1130534 T>A SNP genotype with patient characteristics.

Subjects	*n* = 101	TT	TA	AA	χ^2^	df	*p*-Value
Association with gender							
Males	52	44	6	2	2.29	2	0.15
Females	49	46	2	1			
Association with Age							
Age < 40	41	34	5	2	2.73	2	0.25
Age ≥ 40	60	56	3	1			
Fasting blood glucose							
Glucose ≤ 110 mg/dL	14	8	4	2	17.62	2	0.0001
Glucose > 110 mg/dL	87	82	4	1			
Association with RBS							
RBS ≤ 200 mg/dL	56	51	3	2	1.25	2	0.53
RBS > 200 mg/dL	45	39	5	1			
Association with total cholesterol							
Cholesterol < 200 mg/dL	60	57	1	2	7.94	2	0.0189
Cholesterol > 200 mg/dL	41	33	7	1			
Association with HDL-C							
HDL-C ≤ 55 mg/dL	75	70	3	2	10.44	2	0.005
HDL-C > 55 mg/dL	28	20	5	1			
Association with LDL-C							
LDL ≤ 100 mg/dL	75	71	2	2	11.25	2	0.003
LDL > 100 mg/dL	26	19	6	1			
Association with TG							
TG ≤ 150 mg/dL	64	59	3	2	2.51	2	0.28
TG > 150 mg/dL	37	31	5	1			
Association with HbA1c							
Hb A1c ≤ 6%	31	22	7	2	15.61	2	0.004
HbA1c > 6%	70	68	1	1			
Association with Creatinine							
Creatinine ≤ 1.35 mg/dL	83	68	6	2	0.12	2	0.94
Creatinine > 1.35 mg/dL	18	22	2	1			

RBS—Random blood sugar, HbA1c—glycated hemoglobin, TG—Triglycerides, LDL—C low-density lipoprotein Cholesterol, HDL-C—High-density lipoprotein cholesterol.

**Table 10 jpm-12-00639-t010:** (**A**) Serum methylglyoxal in T2DM patients and controls. (**B**) Serum methylglyoxal in male and female patients. (**C**) Serum methylglyoxal in different age groups in patients.

**(A)**			
	**T2DM Patients (*n* = 80)**	**Controls (*n* = 80)**	***p*-Value ***
[MG] Mean + SD (pg/mL)	258.87 ± 77.10	141.79 ± 33.27	*p* < 0.0001 t-statistic = −5.375
**(B)**			
	267.80 ± 79.82 (males)	243.99 ± 71.48 (females)	*p* = 0.1835 t-statistic = −1.342
**(C)**			
	**Age <40 years (*n* = 30)**	**Age ≥ 40 years (*n* = 50)**	***p*-Value**
Mean + SD	212.31 ± 64.52	271.93 ± 80.90	*p* = 0.0037 t-statistic = 2.990

* *p*-value is calculated by t-test.

## Data Availability

Data used in this study is available upon reasonable request.

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
