# Peer review of "Clinical Implications of Glyoxalase1 Gene Polymorphism and Elevated Levels of the Reactive Metabolite Methylglyoxal in the Susceptibility of Type 2 Diabetes Mellitus in the Patients from Asir and Tabuk Regions of Saudi Arabia"

_jpm, 2022, doi:10.3390/jpm12040639_

Round 1
Reviewer 1 Report
This is an interesting study investigating the association of abnormalities in the glyoxylase enzyme system with type 2 diabetes among among patients in Asir and Tabuk regions of Saudi Arabia. Blood samples were obtained from both healthy and TD2M patients and subsequently, the authors measured the methylglyoxal levels and determined the GLO1 genotypes. Overall, the manuscript was written in good English and easy to follow. Some of my comments that could potentially improve the overall quality of the study:
1) A diagram describing the glyoxylase system could be included in the introduction section.
2) A simple map highlighting the Asir and Tabuk regions of Saudi Arabia should be included to give readers an idea on the location of the regions.
3) Figures 3-4: Are there any significance difference between both groups?
4) More discussion should be presented: Could you speculate on how the CA genotype and the C allele of the GLO1 rs4647 A>C could increase susceptibility to T2DM and how the AT genotype and the A allele of the rs1130534 decreases the susceptibility to T2DM?
Author Response
Please check the replies in the PDF file.

Reviewer 2 Report
The paper is interesting, but the sample size is the minimum acceptable for this type of approach.
The document is well structured, but the results and discussion could be improved.
In material and methods, should refer to collection tubes with EDTA and collection tubes for serum, without anticoagulant. Do no refer tubes by cap colours.
The results in table 2, related to HbA1c are exchanged, please adjust.
|
HbA1c<6 % |
31 |
69.30% |
|
HbA1c >6 % |
70 |
30.69% |
Include the legend of abbreviatures of the table 8 (ex: RBS)
The authors should also present the comparison of methylglyoxal concentration (mean± SD) in the age group of, 40 years (group 1) and ≥ 40 years (group 2) in controls, as well as the same analysis for the different genders in controls.
The authors could evaluate the correlation of the presence of rs2736654 (rs4647) A>C SNP genotype distribution and the levels of MG, according to that analysis could be possible to improve the discussion and justify the phrase “This postulation is substantiated by our ELISA data that showed significantly increased concentrations of the MG”
Author Response
The results in table 2, related to HbA1c are exchanged, please adjust.
Response:
Thank you very much. Correction has been done in the table 2 in the manuscript
|
HbA1c <6 % |
31 |
69.30% |
|
HbA1c >6 % |
70 |
30.69% |
Include the legend of abbreviatures of the table 8 (ex: RBS)
Response: -Thank you very much. The needful is done in Table 8.
The authors should also present the comparison of methylglyoxal concentration (mean± SD) in the age group of, 40 years (group 1) and ≥ 40 years (group 2) in controls, as well as the same analysis for the different genders in controls.
Response: Thank you very much. The needful s done.
The authors could evaluate the correlation of the presence of rs2736654 (rs4647) A>C SNP genotype distribution and the levels of MG, according to that analysis could be possible to improve the discussion and justify the phrase “This postulation is substantiated by our ELISA data that showed significantly increased concentrations of the MG”
Response: The putative mechanism is discussed in the manuscript.
Our result showed that CA genotype and the C allele were associated with increased susceptibility to T2DM (Table 4) which is probably due to reduced GLO1 enzyme activity (as a result of structural perturbation) that leads to the accumulation of its cytotoxic substrate MG hereby causing insulin resistance and T2DM. This postulation is substantiated by our ELISA data that showed significantly increased concentrations of the MG in patients as compared to the control group. However, more studies on larger sample sizes in different populations are needed to explain the exact phenomena in details.
Rehab et al., reported that GLO1 C332C SNP was associated with overexpression of GLO1 mRNA and higher enzyme activity in breast cancer patients suggesting its role in the development of breast cancer and its progression from localized to the advanced one. But the exact mechanism has not been discussed
.Rehab S. Abdul-Maksoud,1 Walid SH. Elsayed,2 and Rasha S. Elsayed3The influence of glyoxalase 1 gene polymorphism on its expression at different stages of breast cancer in Egyptian women. Genes Cancer. 2017 Nov; 8(11-12): 799–807.
Round 2
Reviewer 2 Report
.